# Adaptation and Psychometric Validation of the Spanish Version of Two Measures of Agentic Engagement

**DOI:** 10.3390/bs15111545

**Published:** 2025-11-13

**Authors:** Estefania Guerrero, Georgina Guilera, Alba Aza, Juana Gómez-Benito, Maite Barrios

**Affiliations:** 1Department of Social Psychology and Quantitative Psychology, University of Barcelona, 08035 Barcelona, Spain; estefania.guerrero@ub.edu (E.G.); gguilera@ub.edu (G.G.); juanagomez@ub.edu (J.G.-B.); 2Institute of Neurosciences, University of Barcelona, 08035 Barcelona, Spain; 3Department of Personality, Evaluation and Psychological Treatments, University of Salamanca, 37005 Salamanca, Spain; azhernandez@usal.es

**Keywords:** higher education, students, agentic engagement, assessment, validity, reliability

## Abstract

This study focuses on the adaptation and psychometric validation of the Spanish versions of the Agentic Engagement Scale (AES) and the Enlarged Version of the Agentic Engagement Scale (EVAES) in Spanish undergraduate students. Agentic engagement, denoting the educational insights provided by students during classroom instruction, has garnered attention for its potential impact on teaching and learning. Reeve initially developed the AES in 2013, which Mameli and Passini extended from 5 to 10 items in 2019 to create the EVAES. In a sample of 278 undergraduate students, this study thoroughly examined various psychometric properties, including item response distribution, factor structure, internal consistency, test–retest reliability, and validity evidence. The results consistently demonstrated high item-total correlations, a one-factor structure, good internal consistency, and satisfactory test–retest reliability for both scales. Additionally, the study established validity evidence through positive and significant correlations between agentic engagement and academic engagement, as well as with personality traits such as extraversion, agreeableness, conscientiousness, emotional stability, and openness to experience. In conclusion, the Spanish versions of the AES and EVAES emerge as promising tools for assessing agentic engagement among Spanish-speaking undergraduate students, offering valuable insights into their active participation and contributions to the instructional process.

## 1. Introduction

Academic engagement, characterized as the active participation of the student in the learning process, is a multi-dimensional phenomenon ([21]; [49]). Recent studies, considering two theoretical perspectives (i.e., European and American), have identified at least seven dimensions that constitute engagement ([45]). The European model, rooted in organizational psychology, comprehends three dimensions of engagement: (1) vigor, characterized by high levels of energy, (2) dedication, marked by a sense of significance, enthusiasm, inspiration, pride, and challenge, and (3) absorption, characterized by being fully concentrated and deeply engrossed in one’s work ([39]). Meanwhile, the American model, which originated in the educational context, encompasses (1) behavioral engagement, which refers to observable behaviors in the learning process and task completion, (2) emotional engagement, which indicates the emotional response and affective connections within the educational setting, (3) cognitive engagement, which relates to the cognitive effort to learn, and (4) agentic engagement, which involves constructive contributions from the student to classroom instruction ([21]; [34]; [50]).

In recent decades, educational reforms inspired by the European Higher Education Area (EHEA) have emphasized student-centered and competency-based approaches that promote active learning and student participation in instructional processes ([3]). In this context, agentic engagement plays a pivotal role because it fosters students’ proactive involvement in their education, encouraging them to take ownership of their learning experiences and contribute meaningfully to the educational process.

### 1.1. Conceptual Framework

Within higher education, agentic engagement refers to the educational insights and information that students provide during the instructional process. It pertains to active involvement in activities that align with the objectives and intentions of the educators ([20]), representing a unique and proactive form of engagement that encourages ongoing interchanges of ideas between the teacher and the student ([32]). It involves the most recurrent ways in which students positively take ownership of their learning experience, including behaviors such as asking questions, expressing preferences, and providing suggestions ([34]).

Three interconnected factors influence agentic engagement: (1) individual, such as motivation, self-efficacy, and personality traits (e.g., extraversion, conscientiousness); (2) institutional, including the learning environment, peer interactions, and teachers’ autonomy support; and (3) societal, such as educational policies and university admission systems ([15]; [43]). All of these collectively shape students’ willingness and capacity to engage agentically.

Agentic engagement is theoretically grounded in the broader framework of student agency and academic engagement, which emphasizes students’ active role in setting goals, monitoring progress, and adjusting strategies to achieve desired outcomes ([18]; [30]). Moreover, student agency has been identified as a key feature of feedback practices during instruction, for example, giving a broader perspective to analyze context, interactions, and material objects ([27]), aligning with previous studies that advocate incorporating social context ([9]) and autonomy support as essential elements in education ([19]).

### 1.2. Literature Review

Moderate correlations have been observed between engagement and positive contributions to student outcomes, such as academic achievement ([17]) and motivation ([34]; [35]). Furthermore, previous research has revealed positive relationships between students’ behavioral, cognitive, and emotional engagement and the big five personality traits (such as agreeableness, conscientiousness, and openness to experience), with studies also linking extraversion specifically to behavioral engagement ([31]). However, more evidence is needed regarding the relationship between these personality traits and agentic engagement.

For agentic engagement, several studies have examined its correlation with motivation-related characteristics in students, such as self-efficacy, self-concept, personality and interests, and epistemic beliefs ([10]). A proactive personality, which reflects an individual’s inclination to dynamically influence their environment, has been associated with agentic engagement ([25]). In this sense, there is a tendency for students to believe that agentic engagement strategies are more frequently used by individuals who are sociable and extroverted ([51]).

Student agency has been recognized as a significant factor in enhancing learning outcomes ([32]; [43]; [49]). [32] ([32]) found that agentic engagement not only predicts academic achievement, but also contributes to a unique and explained variation in student achievement that cannot be elucidated by the other dimensions of engagement (e.g., behavioral, emotional, and cognitive engagement), since students with high agentic engagement are able to request additional support from teachers if they need it, compared to other types of engagement ([15]). Indeed, [33] ([33]) have underscored the importance of including agentic engagement in the broader conceptualization of engagement to study achievement and academic progress.

### 1.3. Measurement Instruments of Agentic Engagement

Agentic engagement has primarily been evaluated using the 5-item Agentic Engagement Scale (AES) ([10]). The initial version of the AES was developed and validated for Taiwanese secondary school students by [34] ([34]). Later, [32] ([32]) revised the AES items to reflect not only the individual student’s contribution, but also the contextual elements and interactions in class, validating it for South Korean university students. In both cases, the scale showed a unifactorial structure and adequate internal consistency.

Additionally, the instrument has been translated into different languages, specifically Turkish ([46]) and Italian ([21]), for secondary school students, which has further corroborated the unidimensional structure and adequate reliability of the scale. Within the Spanish context, recent studies have validated the AES in secondary school students ([6]) and primary school students ([26]). Both Spanish versions exhibit a unifactorial structure and adequate reliability. However, the generalization of these findings has not yet been performed for university student populations.

In 2019, Mameli and Passini, based on Reeve’s 2013 version, expanded the AES from five to 10 items to provide a more comprehensive measure of agentic engagement by incorporating a broader range of proactive contributions from students ([22]). Specifically, the 10-item scale was developed to consider the instances of student opposition to their instructor and classmates ([51]). The added items take into account the cathartic potential of the students’ resistance in class and their interactions with their peers ([22]). The resulting Enlarged Version of the Agentic Engagement Scale (EVAES) has been validated for Italian secondary school student populations, showing a unifactorial structure and improved internal consistency compared to the previous versions of the AES. However, while a novel agentic engagement scale has been recently developed in the context of massive open online courses ([13]), there is a lack of validated instruments in the context of university education.

### 1.4. The Present Study

Despite the growing attention to student agency in higher education and the potential impact of agentic engagement on learning outcomes ([34]; [35]; [43]), there remains a lack of validated instruments for assessing engagement among Spanish university students.

Spanish universities, which operate under the framework of the EHEA, are characterized by lecture-based instruction complemented by seminars and group assignments, with a growing emphasis on student-centered and competency-based learning approaches ([5]). However, traditional hierarchical dynamics between lecturers and students often persist, leading to classroom environments where students tend to adopt a more receptive and less participatory role during instruction. In this cultural and pedagogical context, exploring agentic engagement becomes particularly relevant, as it allows for the examination of how Spanish undergraduate students express agency and proactive behaviors in educational systems transitioning from lecture-centered to student-centered paradigms ([1]; [32]).

This shortage of targeted measurement instruments limits both theoretical understanding and practical applications in educational contexts. Therefore, the present study aimed to address this limitation by adapting and performing a psychometric validation of the Spanish versions of the AES ([32]) and the EVAES ([22]) for Spanish university students. Specifically, it focused on analyzing the item response distribution, factor structure, internal consistency, and test–retest reliability of both the AES and EVAES. Additionally, this study aimed to examine the relationships of the AES and EVAES with an established measure of academic engagement, as well as with academic achievement, and the Big Five personality traits. Exploring these associations allows for the assessment of convergent validity and provides insight into how agentic engagement operates within a broader network of motivational and dispositional factors.

Based on these premises, the study pursued the following objectives: (1) to adapt and psychometrically validate the Spanish versions of the AES and EVAES; and (2) to examine the associations between agentic engagement, academic engagement, academic achievement, and personality traits. We hypothesized that (a) both scales would exhibit a unidimensional factor structure and yield scores with adequate reliability ([22]; [32]); (b) scores on the AES and EVAES would correlate positively with academic engagement and with the Big Five personality traits, particularly extraversion ([25]; [34]; [51]); and (c) both scales would be positively associated with academic achievement ([43]; [49]).

## 2. Method

### 2.1. Participants

The study sample comprised 278 undergraduate students, including 166 women (59.7%) and 112 men (40.3%). The ages of the participants ranged from 18 to 56 years, with a mean of 21.1 (SD = 4.60). Regarding educational background, the majority were enrolled in Health Sciences courses such as medicine, nursing, or psychology (n = 154, 55.4%), or in Social Sciences such as education, business, or law (n = 84, 30.2%). In terms of marital status, the majority of participants were single (70.9%, n = 197), followed by 28.4% who were married or living with a partner (n = 79). Regarding living arrangements, most participants lived with their family of origin (71.4%, n = 206), while 14.4% (n = 40) lived with friends. Table 1 shows the demographic data of the participants. A total of 62 students completed the test–retest phase, of whom 47 (75.8%) were women.

### 2.2. Instruments

#### 2.2.1. The Agentic Engagement Scale (AES; [32])

The AES ([32]) includes five items that measure the dialectical and transactional participation of students in the classroom (e.g., “I defend my opinions even if they are not in line with those of my classmates”). Item responses use a 7-point Likert scale, ranging from 1 (completely disagree) to 7 (completely agree). The total score ranges from 5 to 35. A higher score indicates elevated levels of agentic engagement. The original version of the AES showed an adequate Cronbach’s alpha (α = 0.84) ([32]).

#### 2.2.2. The Enlarged Version of the Agentic Engagement Scale (EVAES; [22])

The EVAES ([22]) includes 10 items that cover a variety of proactive student contributions (e.g., “If I don’t agree with a teacher’s statement, I tell him/her”). [22] ([22]) based the EVAES on the 5-item scale of [32] ([32]) and added five new items that reflected the facets of agency omitted in the original version, including the interactions and exchanges initiated by students with their peers, students’ opposition in the classroom, and students’ original contributions. Item responses use a 7-point Likert scale, ranging from 1 (completely disagree) to 7 (completely agree). The total score ranges from 10 to 70. A higher score indicates elevated levels of agentic engagement. The original version of the 10-item EVAES showed an adequate Cronbach’s alpha (α = 0.85) and McDonald’s omega coefficient (ω = 0.86) ([22]).

#### 2.2.3. The Mini International Personality Item Pool-Five-Factor Model-Positively Worded Scale (Mini-IPIP-PW; [8]; Spanish Version by [24])

This instrument assesses the big five personality traits: extraversion (e.g., “I talk to a lot of different people at parties”), agreeableness (e.g., “I am interested in other people’s problems”), conscientiousness (e.g., “I often put things back in their proper place”), emotional stability (e.g., “I have few mood swings”), and openness to experience (e.g., “I am interested in abstract ideas”). It includes 20 items that are scored on a 5-point Likert scale, ranging from how well each statement describes them from 1 (not at all) to 5 (completely) to assess each personality trait. The Spanish Mini-IPIP-PW scale has demonstrated validity evidence and adequate reliability (≥0.90), as well as convincing correlations with engagement ([24]).

#### 2.2.4. The Utrecht Work Engagement Scale for Students (UWES-9S; [38]; Spanish Version by [40])

The UWES-9S is a 9-item instrument originally developed to assess work engagement, which was later adapted to the educational context to measure academic engagement, and comprising three factors: vigor (e.g., “When I’m doing my work as a student, I feel I am bursting with energy”), dedication (e.g., “I am enthusiastic about my studies”), and absorption (e.g., “I feel happy when I am studying intensely”). The items are scored on a 7-point Likert scale, ranging from 0 (never) to 6 (always). The total score ranges from 0 to 54. A higher score indicates an elevated level of academic engagement. The three correlated factors of the UWES-9S have been confirmed in the Spanish version, with the overall scale showing adequate internal consistency (α = 0.84) ([40]).

#### 2.2.5. Sociodemographic Questionnaire

An ad-hoc questionnaire was administered to collect sociodemographic data, including information on the participants’ gender, age, and admission grade point average (GPA).

### 2.3. Procedure

Two researchers independently translated and adapted the items of the AES and EVAES into Spanish. Both versions were reconciled to identify any potential inconsistencies and improve the accuracy of the adaptation. After this, the research team conducted multiple meetings to review and refine the items. The Spanish versions of the AES and EVAES are available in Table A1 in the Appendix A.

Data collection was conducted in two phases. The first phase, which employed a snowball sampling strategy initiated through convenience sampling, took place between October–November 2022 and again between October–November 2023. During this phase, invitations to participate in the study were distributed to undergraduate students via their virtual campus platforms. The second phase was conducted from February to April 2023. In this phase, the questionnaire was disseminated through students’ social networks across Spain, and additional university professors were encouraged to share the questionnaire link with their students. To encourage participation, participants were offered the chance to win an Amazon Gift Card of 50 euros as an incentive. Additionally, for test–retest purposes, participants were invited to complete the questionnaire a second time. Those who agreed provided an email address and later received a follow-up link one month after their initial participation, with a one-week window to respond.

The questionnaires and sociodemographic data were collected through the Qualtrics platform (https://www.qualtrics.com). The AES and EVAES instructions explicitly asked participants to respond based on how they generally behave in class, without referring to any specific course. To participate in the study, individuals had to be enrolled in an undergraduate degree program and have proficiency in Spanish. The participants were fully informed of the voluntary nature, anonymity, and confidentiality of the study and were required to provide online informed consent before participating. The study was conducted in accordance with the Declaration of Helsinki ([47]) and received ethical approval from the bioethics committee of the University of Barcelona (IRB00003099).

### 2.4. Data Analysis

Descriptive statistics of the AES and EVAES were obtained at the item level, including the mean, standard deviation, percentage of item endorsement, skewness, and kurtosis coefficients, as well as the corrected item-total correlations. The skewness and kurtosis coefficient values that were considered acceptable were those that fell within the range [−2, +2] ([2]).

The dimensionality of the AES and EVAES was evaluated by confirmatory factor analysis (CFA) employing the weighted least squares means and variance adjusted (WLSMV) estimation, which is appropriate for ordinal items. Given the absence of clear guidelines for applying fit indices to the analysis of ordered categorical variables ([42]; [48]), a two-index combination strategy for assessing the goodness of fit was employed ([12]). Based on evidence suggesting the robustness of the standardized root mean square residual (SRMR) among different estimation methods ([42]), the SRMR was used in combination with the comparative fit index (CFI), with an SRMR ≤ 0.06 and CFI ≥ 0.95 suggesting a good model fit ([11]; [41]).

Scale-level descriptive statistics were calculated (i.e., mean, standard deviation, skewness, and kurtosis coefficients) for the AES, EVAES, UWES-9S, and Mini-IPIP-PW. Furthermore, the internal consistency of these scales was examined using Cronbach’s alpha (α) and McDonald’s omega (ω) coefficients. A minimum alpha value of 0.70 served as the threshold for a reliable measure, as recommended by [28] ([28]).

Validity evidence, based on the relationships between the AES/EVAES and the other measures, was assessed with Pearson correlation coefficients that correlated the AES/EVAES total score with the UWES-S9, admission GPA, and Mini-IPIP-PW trait scores. The coefficients were interpreted according to [4]’s ([4]) criteria (i.e., 0.10 is weak, 0.30 is moderate, and 0.50 is strong).

Lastly, to explore the test–retest reliability, the intraclass correlation coefficient (i.e., single measures of absolute agreement) was calculated. Values above 0.50 are considered poor, those between 0.50 and 0.75 are considered moderate, those between 0.75 and 0.90 are considered good, and those greater than 0.90 are considered excellent ([14]). Mann–Whitney U tests for independent samples were conducted to examine potential group differences in baseline engagement scores between participants who completed the retest and those who did not. No statistically significant differences were observed.

The statistical analyses were performed using IBM SPSS Statistics (version 28) and the R statistical console (version 4.3) with the lavaan package ([36]). 

## 3. Findings

### 3.1. Item-Level Descriptive Statistics

Table 2 shows the distribution of the AES and EVAES item scores. All items covered the full-scale range (i.e., 1–7). The items of both scales had acceptable normality values, with skewness and kurtosis coefficients between −2 and +2. Of particular note, item 7 of the EVAES (“During classes, I raise new questions or topics for discussion”) concentrated the largest percentage of responses at option 1 (completely disagree), showing a floor effect. All the corrected item-total correlations were above 0.50 for both the AES and EVAES.

### 3.2. Factor Structure

The unidimensional model demonstrated a good fit for both the AES data (CFI = 0.98, SRMR = 0.04) and the EVAES data (CFI = 0.95, SRMR = 0.06).

Figure 1 shows the path diagram of the one-factor model of both the AES (a) and the EVAES (b). All factor loadings were high and statistically significant. The values ranged from 0.76 to 0.87 for the AES and from 0.56 to 0.86 for the EVAES.

### 3.3. Internal Consistency and Temporal Stability

Table 3 displays the descriptive statistics of AES and EVAES scores (i.e., mean, standard deviation, skewness, and kurtosis) and their internal consistency coefficients.

Cronbach’s alpha (α) and McDonald’s omega (ω) coefficients indicated that the AES and EVAES scores demonstrated adequate internal consistency. The coefficients were higher for the EVAES than for the AES, likely due to the larger number of items included in the EVAES.

Intraclass correlation coefficients for the test–retest reliability (i.e., temporal stability) were 0.68 (*p* < 0.001) and 0.71 (*p* < 0.001) for the AES and EVAES, respectively, suggesting moderately good test–retest reliability for both instruments.

### 3.4. Validity Based on Correlations with Other Variables

Table 3 shows the correlations of the AES and EVAES scores with the other measures of engagement, academic achievement, and personality traits, along with the descriptive statistics of these measures and their internal consistency coefficients.

The total scores for the AES and EVAES showed moderate associations with the UWES-9S (*r* = 0.29 and *r* = 0.32, respectively). Regarding the relationship between the AES and EVAES scores and personality traits, all factors showed significant associations. For example, positive moderate correlations were observed with extraversion (*r* = 0.28 with AES and *r* = 0.30 with EVAES) and positive weak to moderate correlations were noted with the rest of the personality traits, being the most noteworthy those obtained with the openness to experience domain (*r* = 0.17 with AES and *r* = 0.20 with EVAES). However, the correlation between the AES or EVAES score and academic achievement was very low.

## 4. Discussion

By leveraging their agency and initiative, students actively personalize and enhance the quality of their learning environments. This enhancement boosts teacher support for students, students’ motivation, interest, and learning. Considering these benefits, this study aimed to produce the Spanish versions of the AES and EVAES and to conduct a psychometric validation to determine their applicability among Spanish university students. This study is the first to validate both instruments for university students. Previous studies have primarily focused on primary ([26]) and secondary school students ([6]; [21]; [46]). The Spanish versions of both instruments show promise in measuring agentic engagement in university students. Our findings confirmed their good psychometric properties, specifically high item-total correlations, an adequate one-factor structure, convincing internal consistency, adequate test–retest reliability, and validity evidence based on the relationship with academic engagement and personality traits.

The item-level descriptive statistics of the Spanish versions of the AES and EVAES showed distributional properties like those reported for the original versions ([32]). It is worth noting that Item 7 of the EVAES (“During classes, I raise new questions or topics for discussion”) showed a floor effect, with responses predominantly clustered at the lower end of the scale. This may suggest that actively contributing new ideas during class poses a significant challenge for students within the instructional context ([34]).

Furthermore, the present investigation revealed that the one-factor structure of both the AES ([32]) and the EVAES ([22]) showed a good model fit and high factor loadings of the items. The results for the AES are consistent with those of the validation studies conducted in Spanish primary and secondary school students ([6]; [26]), thus bolstering confidence in replicating this in larger sample sizes.

The internal consistency observed aligns with that of the original versions of both scales ([22]; [32]). The present study noted a higher internal consistency in the Spanish version of the EVAES compared to the AES, similar to previous reports using the original scales ([22]). As for the Spanish version of the AES, its internal consistency was in line with that reported in populations of secondary school students ([6]) and primary school students ([26]). In summary, the scores of both scales appear to be dependable for measuring agentic engagement in Spanish university students. In addition, this study presents evidence of the test–retest reliability of the Spanish versions of the AES and EVAES that has not been previously reported in the validation studies of the original scales ([22]; [32]). In this sense, the findings support the temporal stability of the scores provided by both scales to measure agentic engagement.

The results showed the expected correlation between the AES/EVAES score and academic engagement, as agentic engagement tends to be related to other types of engagement ([34]). Concerning the relationship between the AES/EVAES score and personality traits, a positive relationship was observed for all the traits. Since agentic engagement is often related to a proactive personality (i.e., the yearning to enthusiastically transform their environment) ([25]), traits such as extraversion, agreeableness, conscientiousness, emotional stability, and openness to experience were expected to correlate with the AES/EVAES total scores. This aligns with prior research findings indicating that individuals who are sociable and extroverted tend to employ agentic engagement strategies more often ([51]). To the best of our knowledge, this is the first study to provide evidence on the relationship between agentic engagement and the big five personality traits.

Although average grades may appear to be biased ([7]), they remain the most widely used institutional metric for assessing student academic performance ([17]). The present study revealed that there was no significant correlation between the AES/EVAES score and academic performance, as assessed by the admission GPA scores, indicating a very weak relationship. Several possible factors could explain this finding. First, academic achievement is influenced by a multitude of factors, including the specific characteristics of the university program (e.g., specialization and program experience). This complexity might account for the variations observed in the relationship between academic achievement and other variables across different samples, as previously noted by [7] ([7]). Second, considering that the degree of academic engagement may vary throughout a study period ([25]), the temporal discrepancies between the GPA reported (i.e., past academic achievement) and the agentic engagement score (i.e., at the time of completing the questionnaire) must be considered. Third, there is a possibility that self-reported GPA scores can be overrated or underrated ([16]). These three possibilities may explain the results of the correlation between the GPA and agentic engagement in this study. While some studies have emphasized the relationship between engagement and achievement ([17]; [37]), the findings of the present study are consistent with those of others that have not found this association ([22]).

### 4.1. Limitations and Future Research

This study has some limitations that need to be considered. First, the sample was obtained through a convenience-based snowball sampling strategy and consisted exclusively of students enrolled in undergraduate programs. These constrain the generalizability of the findings to the broader higher education population. Postgraduate and doctoral students may experience different levels of autonomy, motivation, and engagement shaped by their academic maturity and disciplinary contexts. Future research should therefore explore agentic engagement among more diverse student populations, including master’s and doctoral programs. Such studies would provide a more comprehensive understanding of how agentic engagement manifests across the full spectrum of higher education experiences. In addition, the use of incentives may have introduced self-selection biases that could further affect representativeness. Moreover, the participants who completed the test–retest phase included a slightly higher proportion of women, which should be considered when interpreting the test–retest results. Future studies would benefit from stratified sampling designs to ensure adequate sample sizes for conducting measurement invariance analyses across academic and sociodemographic variables (e.g., disciplines, academic level, and gender). Additionally, it should examine whether the AES and the EVAES function equivalently across cultures (e.g., English vs. Spanish versions).

Second, the results relied on students’ self-reported data, which may have led to discrepancies, particularly in the GPA variable ([16]). In this regard, future studies could reduce potential reporting bias by obtaining academic performance data directly from institutional records or instructors ([43]). Further research on the relationship between the Spanish versions of the AES and EVAES and objective academic outcomes would also provide valuable insights.

Finally, future research could examine the predictive potential of these scales in relation to academic success, dropout, and post-graduation job performance, as well as employ longitudinal designs to track changes in agentic engagement across different stages of academic progression. Furthermore, exploring how personality traits or dispositional factors predispose individuals to higher levels of agentic engagement would provide valuable evidence for the development of targeted interventions and educational strategies.

### 4.2. Theoretical and Practical Implications

The results of this study are particularly relevant for higher education programs that share comparable disciplinary, institutional, and cultural characteristics with the Spanish context. Accordingly, our findings may guide strategies to foster agentic engagement in similar educational settings. This enhances the applicability of the results and supports their use in designing pedagogical interventions aimed at promoting student agency across diverse yet structurally analogous higher education contexts.

Despite the limitations of this study, the results are promising and have important implications for both theory and practice. This study provides evidence of the psychometric properties of the Spanish versions of the AES and EVAES, offering two alternative instruments for measuring agentic engagement in Spanish undergraduate students.

This study supports the adequacy of both scales for use in Spanish undergraduate students. The results are consistent with those of the studies in the existing literature, which define agentic engagement as unidimensional ([32]), thus contributing to the theoretical understanding of it. Moreover, having different language versions of the scales may help in the comparison of agentic engagement between cultures (i.e., cross-cultural engagement). While student agency lies at the core of ongoing educational innovations worldwide ([29]), the validated scales can be used to guide future research in Spanish-speaking contexts, including studies on educators’ perceptions and evaluations of student agency as well as on the circumstances in which this concept emerges as a crucial determinant of academic achievement ([20]; [33]). Moreover, integrating the evaluation of agentic engagement into theoretical research may unveil novel insights into agency, thereby enriching pertinent domains such as self-formation ([23]) and engagement theory ([44]).

Since identifying agentic engagement is essential for studying co-creation between teachers and students ([10]), the validation of both scales has implications for university education, contributing to the comprehension of the multidimensional nature of academic engagement ([45]). Furthermore, considering the significance of student agency in self-regulated learning ([30]), feedback practices during instruction ([27]), and the potential of agentic engagement to prevent school failure and dropout ([35]), the assessment of the programs and interventions that aim to support these processes in university students would be valuable. In this sense, this study provides evidence of the psychometric properties of the AES and EVAES that can be used to cover the research gap caused by the lack of valid instruments measuring agentic engagement in Spanish university students.

## 5. Conclusions

In this study, the Spanish versions of the AES and EVAES demonstrate promising potential as instruments for assessing agentic engagement. They show good psychometric properties in terms of high item-total correlations, an adequate unifactorial structure, convincing internal consistency, adequate test–retest reliability, and positive correlations with academic engagement and personality traits (i.e., extraversion, agreeableness, conscientiousness, emotional stability, and openness to experience). Agentic engagement emerges as a well-defined dimension of engagement that provides relevant information on how students engage themselves in the flow of classroom instruction. The Spanish versions of the AES and EVAES can be used for this purpose, depending on the specific application conditions (e.g., time or purpose). In particular, the AES was developed to capture how students typically participate in classroom activities, while the EVAES was developed to account for students’ resistance towards their teacher and fellow peers ([20]). Therefore, the EVAES can be used in studies interested in a more comprehensive conceptualization of agentic aspects, including original student contributions, student opposition in the classroom, and transactional contributions with peers, while the AES can be useful for educational contexts with short study periods where several variables are evaluated at once. While the present study offers valuable insights into agentic engagement among undergraduate students, its findings should be interpreted with caution given the limited representativeness of the sample. Expanding future research to include postgraduate and doctoral students from various disciplines would allow for a more nuanced examination of how academic level shape students’ proactive engagement in learning.

## Figures and Tables

**Figure 1 behavsci-15-01545-f001:**
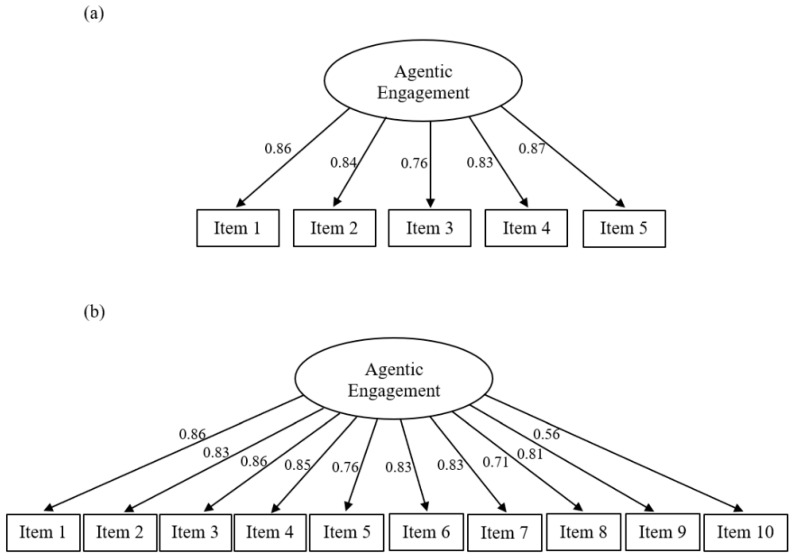
Path diagrams of the (**a**) Agentic Engagement Scale (AES) and the (**b**) Enlarged Version of the Agentic Engagement Scale (EVAES).

**Table 1 behavsci-15-01545-t001:** Sociodemographic data.

Variable	n	%
Gender		
Female	166	59.7
Male	112	40.3
Educational Background		
Health sciences	154	55.4
Social sciences	84	30.2
Engineering	17	6.1
Arts and Humanities	9	3.2
Natural Sciences	8	2.9
Mixed background	4	1.4
No data	2	0.7
Marital status		
Single	197	70.9
Married/with a partner	79	28.4
Separated/divorced	2	0.7
Living arrangements		
Family of origin	206	71.4
Partner or own family	10	3.6
Friends	40	14.4
Alone	7	2.5
Residential institution	9	3.2
Other	6	2.2

**Table 2 behavsci-15-01545-t002:** Item-level descriptive statistics of the AES and EVAES.

Item *	M	SD	PIE (%)							SK	K	r_jx_
			1	2	3	4	5	6	7			
I1 (I3)	3.44	1.85	17.3	24.5	10.8	13.7	16.5	12.6	4.7	0.26	−1.20	0.80 (0.75)
I2	3.08	1.79	21.6	28.4	13.7	9.4	14.7	8.6	3.6	0.56	−0.88	0.77
I3 (I1)	3.32	1.80	17.6	24.1	16.5	11.2	16.2	9.7	4.7	0.39	−1.0	0.79 (0.75)
I4 (I2)	3.18	1.77	20.9	25.6	11.2	13.7	17.6	7.9	3.2	0.38	−1.05	0.78 (0.77)
I5	3.39	1.84	18.3	23.0	12.2	14.4	16.5	10.4	5.0	0.29	−1.11	0.66
I6	3.01	1.67	22.3	24.5	16.5	15.1	12.9	5.8	2.9	0.54	−0.67	0.74
I7	2.80	1.70	29.1	25.9	11.5	12.6	12.9	5.8	2.2	0.66	−0.70	0.75
I8 (I5)	3.54	1.84	18.7	17.6	12.9	12.9	21.6	11.9	4.3	0.08	−1.23	0.63 (0.68)
I9 (I4)	3.62	1.98	18.3	20.9	10.8	11.2	17.6	12.2	9.0	0.19	−1.28	0.74 (0.75)
I10	4.59	1.94	9.7	11.2	6.5	12.9	21.2	19.4	19.1	−0.49	−0.93	0.53

Note. I = item; M = mean; SD = standard deviation; PIE (%) = percentage of item endorsement. Each statement is rated on a 7-point Likert-type scale. SK = skewness; K = kurtosis coefficient. r_jx_ = corrected item-total correlation. * Parentheses include items corresponding to the AES scale.

**Table 3 behavsci-15-01545-t003:** Descriptive statistics, internal consistency coefficients, and correlations of the AES and EVAES scores with engagement, academic achievement, and personality traits.

	M	SD	SK	K	α	ω	*r* (AES)	*r* (EVAES)
AES	17.1	7.75	0.16	−0.92	0.89	0.90	-	0.96 ***
EVAES	34.0	14.1	0.17	−0.86	0.93	0.93	0.96 ***	-
UWES-9S	38.7	11.7	−0.30	−0.36	0.94	0.94	0.32 ***	0.29 ***
Admission GPA	7.88	1.20	−0.57	1.25	-	-	−0.02	−0.07
Mini-IPIP-PW								
Extraversion	10.8	3.6	0.18	−0.58	0.73	0.73	0.28 ***	0.30 ***
Agreeableness	15.3	3.1	−0.46	−0.35	0.83	0.83	0.16 **	0.14 *
Conscientiousness	13.1	3.5	−0.15	−0.52	0.81	0.82	0.16 **	0.14 *
Emotional stability	10.4	3.1	0.19	−0.20	0.66	0.67	0.13 *	0.12 *
Openness to experience	13.4	4.0	−0.14	−0.80	0.86	0.86	0.17 **	0.20 ***

Note. M = mean; SD = standard deviation; SK = skewness; K = kurtosis coefficient; α = Cronbach’s alpha; ω = McDonalds’ omega; *r* = Pearson correlation. * *p* < 0.05; ** *p* < 0.01; *** *p* < 0.001 (two-tailed).

## Data Availability

Data could be obtained by contacting the corresponding author.

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
