# Peer review of "Adaptation and Psychometric Validation of the Spanish Version of Two Measures of Agentic Engagement"

_behavsci, 2025, doi:10.3390/bs15111545_

Round 1

Reviewer 1 Report

Comments and Suggestions for Authors

This subject is both interesting and relevant, and has attracted the interest of researchers in recent years.

The authors invest in conceptualising the phenomena under study, focusing both on agentic engagement and other related constructs (self-regulated learning, academic performance and personality traits).

The measures for assessing agentic engagement are contextualised, and the purpose and potential impact of validating the measures (AES and EVAES) are explained. The purpose of validation in the Spanish population is unambiguous; however, the authors have not explicitly articulated the use of the alternative measures (a cursory reference is made in the final sentence of the section entitled "The present study"). Provision of clarification on the rationale behind the utilization of the Mini-IPIP-PW and UWES-9S, in addition to the formulation of objectives, is needed, given that the results obtained have a demonstrable impact and reveal comparative and explanatory potential.

In the findings section, the following recommendations are needed with regard to Table 3: firstly, the text immediately following the factorial structure should be inserted immediately before Table 3 (and after Figure 1); secondly, the text following point 3.3 should appear after Table 3; and thirdly, the last sentence of point 3.4 should appear before.

In the final paragraph of the discussion, "some studies" are cited; however, there is only a single reference provided to support each of the statements.

With regard to limitations, we propose a reference to sampling, which may be considered a limitation, in addition to the fact that there were incentives and this was not mentioned in the limitations.

Author Response

Dear Editor and Reviewers,

We would like to sincerely thank you for the time and effort you have dedicated to reviewing our manuscript. We are extremely grateful for the thoughtful and constructive feedback provided, which has been invaluable for improving the quality, clarity, and rigor of our work.

Below, we provide a point-by-point response to each comment. For clarity, the reviewers’ comments are presented in bold, followed by our responses in regular text. All modifications made to the manuscript are highlighted in tracked changes in the revised version.

Reviewer 1

This subject is both interesting and relevant, and has attracted the interest of researchers in recent years.

The authors invest in conceptualising the phenomena under study, focusing both on agentic engagement and other related constructs (self-regulated learning, academic performance and personality traits).

The measures for assessing agentic engagement are contextualised, and the purpose and potential impact of validating the measures (AES and EVAES) are explained. The purpose of validation in the Spanish population is unambiguous; however, the authors have not explicitly articulated the use of the alternative measures (a cursory reference is made in the final sentence of the section entitled "The present study"). Provision of clarification on the rationale behind the utilization of the Mini-IPIP-PW and UWES-9S, in addition to the formulation of objectives, is needed, given that the results obtained have a demonstrable impact and reveal comparative and explanatory potential.

We thank the reviewer for this insightful observation. We have expanded the final paragraph of the “The present study” section to clarify the rationale for including the Mini-IPIP-PW and UWES-9S scales. Specifically, we now explain that these instruments were included to examine AES and EVAES convergent validity (with work engagement) and their relationship with personality traits.

Additionally, we have expanded the objectives of the study at the end of this section. We now explicitly state that one of the aims of this research was to analyze the relationships between agentic engagement, work engagement, and personality traits to provide explanatory insights into how these constructs interrelate in the Spanish university context. We also specify the corresponding hypotheses guiding these analyses.

We hope these changes make the objectives of the study more explicit and theoretically grounded, thereby strengthening the rationale for including these instruments.

In the findings section, the following recommendations are needed with regard to Table 3: firstly, the text immediately following the factorial structure should be inserted immediately before Table 3 (and after Figure 1); secondly, the text following point 3.3 should appear after Table 3; and thirdly, the last sentence of point 3.4 should appear before.

We thank the reviewer for the careful reading and for this valuable suggestion. We have reorganized the placement of the paragraphs as indicated:

  • The paragraph following the description of the factorial structure now precedes Table 3.
  • The text of Section 3.3 follows Table 3.
  • The last sentence of Section 3.4 has been moved as instructed.

In the final paragraph of the discussion, "some studies" are cited; however, there is only a single reference provided to support each of the statements.

We appreciate this observation. We have revised the paragraph to provide an additional reference.

With regard to limitations, we propose a reference to sampling, which may be considered a limitation, in addition to the fact that there were incentives and this was not mentioned in the limitations.

We fully agree. We have added a sentence in the Limitations section acknowledging that the sampling strategy and the use of incentives may limit the generalizability of the findings.

Reviewer 2 Report

Comments and Suggestions for Authors

The manuscript addresses a topic of clear relevance to the field of educational psychology, particularly within the context of higher education, where students’ active engagement is increasingly recognized as a key determinant of learning quality.

To strengthen the methodological rigor, it is recommended that the authors ensure stratification by field of study, thereby enabling measurement invariance analyses across academic disciplines.

The manuscript would further benefit from a more explicit articulation of the specific scientific problem being addressed, accompanied by the formulation of clear research hypotheses.

In addition, the sample size used in the test–retest phase should be reported, together with an assessment of potential systematic attrition (i.e., dropout bias). Please explain how it was ensured that the same participants' data were included in the test-retest phase of the study.

Item 7 of the EVAES (“During classes, I raise new questions or topics for discussion”) appears to exhibit a floor effect, with responses predominantly clustered at “strongly disagree.” It is therefore advisable either to revise the item wording or to provide a contextual explanation for this pattern—possibly related to cultural or classroom interaction norms.

With respect to ethical and procedural aspects, full details of the ethical approval process should be included, particularly if the study is intended for publication in an open-access format.

Finally, a more explicit discussion of the study’s limitations is recommended.

Implementing these refinements would enhance the manuscript’s methodological transparency, reliability, and overall contribution to the literature.

Author Response

Dear Editor and Reviewers,

We would like to sincerely thank you for the time and effort you have dedicated to reviewing our manuscript. We are extremely grateful for the thoughtful and constructive feedback provided, which has been invaluable for improving the quality, clarity, and rigor of our work.

Below, we provide a point-by-point response to each comment. For clarity, the reviewers’ comments are presented in bold, followed by our responses in regular text. All modifications made to the manuscript are highlighted in tracked changes in the revised version.

Reviewer 2

The manuscript addresses a topic of clear relevance to the field of educational psychology, particularly within the context of higher education, where students’ active engagement is increasingly recognized as a key determinant of learning quality.

To strengthen the methodological rigor, it is recommended that the authors ensure stratification by field of study, thereby enabling measurement invariance analyses across academic disciplines.

We appreciate this valuable methodological suggestion. Unfortunately, our sample size within each field of study was insufficient to conduct meaningful measurement invariance analyses. We acknowledge the importance of this approach and note it as a limitation, suggesting that future studies employ stratified sampling to enable such analyses.

The manuscript would further benefit from a more explicit articulation of the specific scientific problem being addressed, accompanied by the formulation of clear research hypotheses.

We thank the reviewer for this excellent suggestion. We have revised the Introduction to make the research problem more explicit and included clear research objectives and hypotheses at the end of the “The present study” section. We hope these changes clarify the scientific problem and provide a stronger theoretical foundation for the study.

In addition, the sample size used in the test–retest phase should be reported, together with an assessment of potential systematic attrition (i.e., dropout bias). Please explain how it was ensured that the same participants' data were included in the test-retest phase of the study.

We thank the reviewer for this important suggestion. The sample for the test–retest phase (n = 62) and potential systematic attrition with respect to sociodemographic characteristics are reported in the first paragraph of the Participants section, and we have included this as a possible limitation regarding the test–retest. In the Data Analysis section, we have added an analysis of group differences on baseline scores of the engagement measures to assess whether participants who completed the retest differed from those who did not. Additionally, at the end of the second paragraph of the Procedure section, we have clarified how it was ensured that the same participants’ data were matched across both assessments using their email addresses.

Item 7 of the EVAES (“During classes, I raise new questions or topics for discussion”) appears to exhibit a floor effect, with responses predominantly clustered at “strongly disagree.” It is therefore advisable either to revise the item wording or to provide a contextual explanation for this pattern—possibly related to cultural or classroom interaction norms.

We thank the reviewer for this pertinent remark. We have added a contextual explanation to the Discussion section, highlighting that this pattern likely reflects cultural and interaction norms in Spanish university classrooms, where students rarely initiate topics during lectures.

With respect to ethical and procedural aspects, full details of the ethical approval process should be included, particularly if the study is intended for publication in an open-access format.

We have added the full name of the ethics committee, the approval number, and confirmation of adherence to institutional protocols in the appropriate section of the manuscript.

Finally, a more explicit discussion of the study’s limitations is recommended.

We thank the reviewer for this valuable suggestion. We have revised and expanded the Limitations section to provide a more explicit discussion of the study’s limitations. These revisions include additional details regarding the sample characteristics (e.g., convenience-based snowball sampling, and incentives), and potential systematic attrition in the test–retest phase (e.g., gender differences). Furthermore, we have highlighted directions for future research to address these limitations.

Implementing these refinements would enhance the manuscript’s methodological transparency, reliability, and overall contribution to the literature.

Reviewer 3 Report

Comments and Suggestions for Authors

This review pertains to the article "Adaptations to Psychometric Validation and the Spanish Version of Two Measures of Agentic Engagement." I complete research in the area of college student success, with a particular focus on mental health. Therefore, this paper was of particular interest to me.  

The purpose of this paper was to adapt two previously used psychometric tests, AES and EVAES, for use in Spanish with undergraduate students. Overall, this paper is written with a clear tone and valuable statistical analysis. Moving forward with this manuscript, several suggestions are listed below. 

Introduction 

The paper lacks a section on a theoretical or conceptual framework. However, conceptual ideas are provided in the fourth paragraph of the introduction. Consider developing this section as a conceptual framework and include a figure to display how these ideas are interconnected.  

Consider using an additional header, such as Literature Review, with the information starting with “Moderate correlations...” in the fifth paragraph of the introduction.  

Tables

It is generally practice to have the columns with numbers centered, as presented in the paper. However, the left-most column is usually left-aligned. It helps with readability. Consider making this change to the tables in the paper. 

Findings 

Consider elaborating on the study's findings. Consider including the differences present between various demographic characteristics.  

Survey: As this paper is written in English, it would be helpful if the survey were also presented in English, in addition to Spanish. 

The paper mentions that the participants also took the Mini-IPIP-PW and the Utrecht Work Engagement Scale. It would be beneficial to include these as artifacts in the appendix as well.  

Overall, the paper is well written. In addition to survey validation, include more findings related to this sample population.  

Author Response

Dear Editor and Reviewers,

We would like to sincerely thank you for the time and effort you have dedicated to reviewing our manuscript. We are extremely grateful for the thoughtful and constructive feedback provided, which has been invaluable for improving the quality, clarity, and rigor of our work.

Below, we provide a point-by-point response to each comment. For clarity, the reviewers’ comments are presented in bold, followed by our responses in regular text. All modifications made to the manuscript are highlighted in tracked changes in the revised version.

Reviewer 3

This review pertains to the article "Adaptations to Psychometric Validation and the Spanish Version of Two Measures of Agentic Engagement." I complete research in the area of college student success, with a particular focus on mental health. Therefore, this paper was of particular interest to me.  

The purpose of this paper was to adapt two previously used psychometric tests, AES and EVAES, for use in Spanish with undergraduate students. Overall, this paper is written with a clear tone and valuable statistical analysis. Moving forward with this manuscript, several suggestions are listed below. 

Introduction 

The paper lacks a section on a theoretical or conceptual framework. However, conceptual ideas are provided in the fourth paragraph of the introduction. Consider developing this section as a conceptual framework and include a figure to display how these ideas are interconnected.  

We thank the reviewer for this helpful suggestion. In response, we have added a new section entitled “Conceptual Framework” to highlight conceptual ideas in the Introduction. Rather than creating a figure, we have expanded the “Conceptual Framework” section to provide a more detailed explanation of how agentic engagement is framed within the present study. This textual elaboration clarifies the relationships among individual, contextual, and motivational factors without requiring a visual representation.

Consider using an additional header, such as Literature Review, with the information starting with “Moderate correlations...” in the fifth paragraph of the introduction.  

We have reorganized the Introduction, adding a “Literature Review” heading before the section starting with “Moderate correlations…”

Tables

It is generally practice to have the columns with numbers centered, as presented in the paper. However, the left-most column is usually left-aligned. It helps with readability. Consider making this change to the tables in the paper. 

We have reformatted all tables (except Table 2) so that the left most column is left-aligned.

Findings 

Consider elaborating on the study's findings. Consider including the differences present between various demographic characteristics.  

Thank you for your comment. We did not compare the scores across sociodemographic variables because it would first be necessary to demonstrate at least scalar invariance for each of these variables, which was not possible due to the limited sample size. However, we have now added a sentence indicating that future studies should address measurement invariance across sociodemographic groups.

Survey: As this paper is written in English, it would be helpful if the survey were also presented in English, in addition to Spanish. 

The paper mentions that the participants also took the Mini-IPIP-PW and the Utrecht Work Engagement Scale. It would be beneficial to include these as artifacts in the appendix as well.  

We have now included the English version of the questionnaires being validated in this study (AES and EVAES), as it directly enhances the clarity and applicability of our work. We have not added the other instruments used (i.e., Mini-IPIP-PW and Utrecht Work Engagement Scale), since their validated English versions are already available in the original publications, which we have duly cited. This approach ensures proper acknowledgment of the original authors and avoids unnecessary duplication of previously published materials.

Overall, the paper is well written. In addition to survey validation, include more findings related to this sample population.  

The information about the study sample provided in Section 2.1 (Participants) within the Method has been expanded. New details regarding the students’ educational level have been added and the description of the data presented in Table 1 has been enhanced.

Round 2

Reviewer 2 Report

Comments and Suggestions for Authors

Thank you for the work undertaken in refining the study report.

Given that the sample is non-representative, it is methodologically appropriate to clearly define the context of the study—specifically, what makes this investigation unique. This includes a detailed description of the case under analysis, such as the institutional, cultural, and academic environment in which the research was conducted.

If the study examines typical cases (e.g., undergraduate students from mid-sized universities), it is advisable to strengthen the purpose of the research by emphasizing its relevance to similar educational settings. Doing so would enhance the study’s contribution to the broader understanding of agentic engagement in higher education.

It is important to note that the majority of participants were undergraduate students, which limits the generalizability of the findings to the entire higher education population. This limitation should be explicitly acknowledged in the discussion and conclusion sections, along with recommendations for future research involving more diverse student groups, including postgraduate and doctoral students across various disciplines.

It would also be appropriate to indicate which module/subject is being engaged in. As you know, learners may be engaged in different subjects in different ways. And this may be influenced by different factors - the content of the subject, the attractiveness of the teaching, the commitment and motivation of the students themselves, etc... The question is, what subject should the students have been thinking about when answering the questions?

Author Response

We would like to thank the reviewer for their careful reading and the suggestions provided to improve the manuscript. We have implemented the suggested changes in the new version of the manuscript, highlighted in yellow. Please find our responses to each comment below. For each comment, we provide: (1) our observations and (2) the location in the text where the changes have been made.

Reviewer comment: Given that the sample is non-representative, it is methodologically appropriate to clearly define the context of the study—specifically, what makes this investigation unique. This includes a detailed description of the case under analysis, such as the institutional, cultural, and academic environment in which the research was conducted.

[Response]: We thank the reviewer for this insightful comment. Following this suggestion, we have expanded the Introduction (page 3, section 1.4. The present study) to include additional information about the higher education context in which the study was conducted, emphasizing the characteristics of Spanish universities and the learning environment of undergraduate students in this setting.

Reviewer comment: If the study examines typical cases (e.g., undergraduate students from mid-sized universities), it is advisable to strengthen the purpose of the research by emphasizing its relevance to similar educational settings. Doing so would enhance the study’s contribution to the broader understanding of agentic engagement in higher education.

[Response]: We appreciate the reviewer’s thoughtful suggestion. In response, we have revised the end of the Discussion (page 11, section 4.2. Theoretical and practical implications) sections to underscore how our findings may contribute to understanding agentic engagement in typical university contexts—particularly in mid-sized, traditional universities where instructional dynamics are evolving toward more participatory models.

Reviewer comment: It is important to note that the majority of participants were undergraduate students, which limits the generalizability of the findings to the entire higher education population. This limitation should be explicitly acknowledged in the discussion and conclusion sections, along with recommendations for future research involving more diverse student groups, including postgraduate and doctoral students across various disciplines.

[Response]: We appreciate the reviewer’s observation and agree that the limitation concerning the sample composition deserves more emphasis. Since our sample consisted exclusively of students enrolled in undergraduate programs, we have addressed how this constrains the generalizability of our findings (Page 10, section 4.1. Limitations and future research). Additionally, we now include clearer recommendations for future studies involving postgraduate and doctoral populations across different disciplines. Corresponding adjustments were also made in the Conclusions section (Page 12).

Reviewer comment: It would also be appropriate to indicate which module/subject is being engaged in. As you know, learners may be engaged in different subjects in different ways. And this may be influenced by different factors - the content of the subject, the attractiveness of the teaching, the commitment and motivation of the students themselves, etc... The question is, what subject should the students have been thinking about when answering the questions?

[Response]: We thank the reviewer for this pertinent observation. The scale’s own instructions explicitly asked participants to respond based on how they generally behave in class, without referring to any specific course (“You will find below a series of statements about your behavior in class. Please indicate the extent to which you identify with each of them. Answer based on how you generally behave in class, without referring to any specific course”). We apologize for not having included these instructions in English in the original submission. We have now added the original instructions (in English) to the Supplementary Material, together with the full set of items and response options, as previously indicated.

Additionally, since the scales do not refer to any specific course but rather assess students’ general behavior in the classroom context, we have emphasized this on page 6, in the Methods section (2.3. Procedure).